# The Importance of the Position of the Nucleus in *Drosophila* Oocyte Development

**DOI:** 10.3390/cells13020201

**Published:** 2024-01-22

**Authors:** Jean-Antoine Lepesant, Fanny Roland-Gosselin, Clémentine Guillemet, Fred Bernard, Antoine Guichet

**Affiliations:** Université Paris Cité, CNRS, Institut Jacques Monod, 75013 Paris, France; jean-antoine.lepesant@ijm.fr (J.-A.L.); fanny.roland-gosselin@ijm.fr (F.R.-G.); clementine.guillemet@gmail.com (C.G.); frederic.bernard@ijm.fr (F.B.)

**Keywords:** oocyte, nucleus, microtubules, oogenesis, *Drosophila*

## Abstract

Oogenesis is a developmental process leading to the formation of an oocyte, a haploid gamete, which upon fertilisation and sperm entry allows the male and the female pronuclei to fuse and give rise to a zygote. In addition to forming a haploid gamete, oogenesis builds up a store of proteins, mRNAs, and organelles in the oocyte needed for the development of the future embryo. In several species, such as *Drosophila*, the polarity axes determinants of the future embryo must be asymmetrically distributed prior to fertilisation. In the *Drosophila* oocyte, the correct positioning of the nucleus is essential for establishing the dorsoventral polarity axis of the future embryo and allowing the meiotic spindles to be positioned in close vicinity to the unique sperm entry point into the oocyte.

## 1. Anatomy and Development of the *Drosophila* Egg Chamber 

A typical ovary comprises approximately 16 ovarioles, each representing an independent egg assembly chain with a tubular organisation. Each ovariole encloses at its tip a structure, the germarium, which is associated with germline and somatic stem cells, whose progeny becomes organized into ovarian follicles or egg chambers. The follicles exit the germarium and continue to develop through a process divided into 14 stages on morphological grounds as they move posteriorly within the ovariole (Figure 1 and Figure 2) [1]. Throughout *Drosophila* oogenesis in adult females, unlike in mammals, germ line stem cells produce a constant supply of new oocytes [2,3].

Following an asymmetric division, a germline stem cell (GSC) produces another stem cell and a cystoblast, which undergoes four incomplete mitotic divisions to generate a cyst of 16 cells connected by cytoplasmic bridges or “ring canals” [10]. Initially, two cells within the cyst with four ring canals initiate premeiotic development with the pairing of homologous chromosomes and the assembly of the synaptonemal complex along the chromosome arms [11]. Then, at stage 1 in the germarium, one of these two cells differentiates into an oocyte, remaining in meiosis, while the 15 other cells of the cyst exit meiosis and eventually endoreplicate their genome to become nurse cells. The oocyte nucleus arrested in prophase of meiosis I, is transcriptionally quiescent or at least poorly active and the chromosomes are condensed in a structure named karyosome. Nurse cells become highly polyploids and provide the oocyte with all types of RNAs, proteins, and cellular organelles [10]. Similarly to mammals, there is a long pause in meiotic prophase, between the pachytene stage where recombination occurs and the metaphase I stage where a meiotic spindle assembles at stage 13 [4]. 

Nurse cells supply to the oocyte is closely linked to the microtubule (MT) cytoskeleton, through a polarized transport from the nurse cells to the oocyte [12,13,14,15]. MT organisation in the oocyte depends on several sources associated or not with the centrosomes [16,17,18]. During the four incomplete divisions that ensure cyst formation, the 16 cells are linked to each other through the ring canals by a cytoplasmic structure known as the fusome, which combines cytoskeleton and vesicles [10]. At the level of the fusome, the spectraplakin Shot, recruits the MT minus end stabilizer Patronin. During cyst formation, asymmetric fusome segregation results in one of the two pro-oocyte cells with four ring canals having more fusome material than the other. Together with the dynein MT motor, Shot and Patronin through an amplification process, create an MT enrichment in the pro-oocyte where the fusome is more abundant, which is required for oocyte specification [19]. Therefore, the selection of the oocyte relies on the formation in the future oocyte of a noncentrosomal microtubule organizing center (ncMTOC) that organises a polarized MT network directing the dynein-dependent transport of cell fate determinants and centrosomes into the pro-oocyte [19,20,21]. 

Centrosomes have a peculiar organisation in the cyst and in the developing oocyte. When the oocyte is determined, the centrosomes migrate from the 15 nurse cells through ncMTs towards the oocyte [19,21]. Because the oocyte is arrested in prophase of meiosis 1, the centrosomes are partially duplicated and between 15 and 32 centrosomes can be observed in the oocyte [18]. During stages 5 to 6, the centrosomes cluster and migrate between the nucleus and the posterior pole of the oocyte where they are active and eventually migrate with the nucleus to an antero-dorsal position [18,22,23]. The meiotic spindle is devoid of centrosomes. This is important because the centrosome of the future zygote is brought by the spermatozoid and the maternal centrosomes, if being maintained, would hinder the formation of the zygote. Hence, in the developing oocyte the centrosomes have disappeared by the end of oogenesis. Elegant experiments have shown the pericentriolar material is first gradually eliminated from the centrosomes in a stepwise manner before the centrioles disappear [24]. 

An important issue is the asymmetric distribution during oogenesis of the determinants of the antero-posterior and dorsoventral polarity axes of the future embryo, which are essential for its segmentation and the formation of the different germ layers. This process relies on the asymmetric localisation of several mRNAs in the oocyte, with those of the *bicoid*, *oskar* (*osk*), and *gurken* (*grk*) genes of particular importance [25]. These mRNAs are transcribed in the nurse cells and transported in a MT-dependent manner into the oocyte through the ring canals (Figure 1D) [26]. Importantly, they are translated only when their transport is completed, ensuring the localisation of the encoded proteins is restricted to the area where the mRNAs are transported and anchored. The *bicoid* mRNA is transported to the anterior pole of the oocyte and encodes a homedomain transcription factor that specifies the anterior of the developing embryo [27,28]. The *osk* mRNA is transported to the posterior pole of the oocyte. Oskar is an RNA scaffold protein that (1) recruits the *nanos* mRNA whose encoded protein controls the posterior segmentation of the embryo and (2) recruits the determinants necessary to specify the future germ cells at the posterior of the embryo [29,30,31,32,33]. The *grk* mRNA is transported close to the oocyte nucleus and relocalises with it when the nucleus migrates to an asymmetric position [34]. The Grk protein is a TGF-alpha ortholog [34]. Upon its local translation, it is secreted toward the nearby follicle cells, where it activates the EGF receptor (EGFr) and triggers several specific differentiation programs in those cells [7]. 

During early oogenesis, before stage 6, the *grk* mRNA is localized close to the nucleus hemisphere facing the posterior of the oocyte, and its translation leads to the activation of the EGFr in about 20 follicle cells adjacent to the oocyte. This induces a differentiation program necessary for building the posterior structures of the eggshell [35,36,37,38] and for the emission, in conjunction with the JAK-STAT signalling pathway, of a signal from those cells, that later triggers a repolarization of the MT network of the oocyte and the start of nucleus migration [37,38,39] (Figure 1C).With the asymmetrical positioning of the nucleus to an antero-dorsal position, the *grk* mRNA, still associated with the nucleus, relocalises in the vicinity of the antero-dorsal follicle cells where its translation triggers a second wave of EGFr activation in a gradient pattern (Figure 1E). This enables the formation of two groups of follicle cells, which forms the dorsal appendages. Through a series of steps, the cells that lack EGFr activation because they do not receive the Grk signal, secrete the ligand triggering the formation of most of the different germ layers of the embryo via the activation of the Toll receptor [7,8]. A crucial step in this process is the asymmetrical positioning of the nucleus, which functions as a symmetry-breaking event for the formation of the dorsoventral axis of the eggshell and of the future embryo. Because the lateral follicle cells are all equivalent before this second wave of Grk signaling, and because there are no markers that predict where the nucleus would move within the oocyte, it is thought the nucleus can migrate towards any position representing the intersection between the anterior and lateral parts of the oocyte. Further support for this view comes from an elegant experiment showing in oocytes with two nuclei, the nuclei migrate to random positions with respect to each other [40].

## 2. Steps in Nucleus Positioning during Oocyte Development

Nuclear positioning occurs between stages 5 and 7, with the nucleus assuming an asymmetric position at later stages [18,22,23,41]. A peculiarity of the oocyte nuclear migration is that it is a three-dimensional process. At stage 5, the oocyte can be assimilated to a truncated cone whose base is in contact with the apical part of the posterior follicular epithelium and the lateral and upper sides in contact with four nurse cells through ring canals and cell-cell contacts. At this stage, the oocyte has an anterior-posterior asymmetry, which is established after the oocyte is determined, such that the plasma membrane facing the posterior follicle cells constitutes the posterior pole of the oocyte and the plasma membrane in contact with the nurse cells the anterior side (Figure 3A,B). At this stage, the dorsoventral polarity is not yet established, this being achieved with the asymmetric positioning of the nucleus as mentioned above. At stage 7, the shape of the oocyte develops into an asymmetric ellipse with the same characteristics for the anterior-posterior polarity. The nucleus is asymmetrically positioned at the intersection of the anterior and posterior plasma membranes, which corresponds to a ring at the edge of the anterior plasma membrane of the oocyte (Figure 3A,B). Importantly, the nucleus can be located at any point on this circumference once its migration is complete as described above. 

The positioning of the oocyte nucleus is a dynamic process during oocyte development. From the germarium to stage 5, the nucleus is positioned close to the anterior plasma membrane (Figure 1A). From stage 5 onwards, the nucleus gradually assumes a central position until stage 6B before migration starts (Figure 3A,B). This centration process relies on the clustering of centrosomes (see below) [23]. This suggests the nucleus has to be centered in the oocyte to migrate. Migration is a relatively slow process taking about two hours for completion [18]. Interestingly, it is a variable process that can be achieved in different ways [18]. In most cases, a biphasic migration pattern was observed. The nucleus hits either the anterior or the lateral plasma membranes before sliding along them to reach its final destination. However, in rare cases (8%), the nucleus migrates directly to the antero-dorsal cortex and does not come into contact with the plasma membrane before its arrival (Figure 3C,D). From stage 7 onward, the nucleus is maintained in its antero-dorsal position until the end of oogenesis and fertilisation. This anchoring is essential to ensure the correct establishment of the dorsoventral polarity of the egg chamber and of the future embryo. Importantly, it has been shown if an asymmetric position of the nucleus was not subsequently maintained, the dorsoventral axis was not determined correctly [42]. Moreover, the antero-dorsal positioning of the nuclei, which is maintained until the end of oogenesis, is likely to be important for the formation of the zygote, as the sperm enters the oocyte from the antero-dorsal side through a channel, the micropyle. Its positioning is controlled during its morphogenesis by the TGF-alpha Grk, which is associated with the oocyte nucleus [9,43] (see below).

### Oocyte Nucleus Positioning and Cytoskeleton

Initial studies evaluating the effect on oocyte development of MT-depolymerising drugs such as colchicine, reported a mispositioning of the oocyte nucleus, highlighting a critical role for MTs [16,44]. These results were further supported by the identification of mutations that affected the dynamics of MTs and impaired nuclear positioning, such as a mutation of the *Drosophila* tubulin-binding cofactor B that enabled the assembly of the alpha tubulin and beta tubulin heterodimer complex [45]. In contrast, although the nucleus is surrounded by a faint ring of actin at mid-oogenesis, latrunculin B, an actin depolymerising drug, at concentrations sufficient to abolish phalloidin-mediated actin detection and impair egg chamber morphogenesis, does not affect nuclear migration (Chemla and Guichet 2024 [46]. Accordingly, a mutation affecting either Cappuccino, a formin, or Chickadee, a profilin [47], both of which are involved in actin assembly in the oocyte, impair MT-based mRNA-associated transport in the oocyte without affecting nuclear positioning. However, it should be noted latrunculin B does not necessarily induce a full depolymerisation of the actin networks. In addition, the *capuccino* and *chickadee* alleles that affect oocyte polarity are not null alleles [47], while a total loss of Chickadee activity prevents oocyte formation [48]. It remains, therefore, to be determined whether actin plays a complementary role to the essential role of MTs.

Prior to its migration the nucleus oscillates around a central position [18]. Given the complexity of the MT network, it was challenging to understand how MTs applied their forces on the nucleus. Depending on whether the MT reorganisation that leads to an inversion of the polarity of the MT networks after the reception of the back signal [16,22,37,38] occurs before or after nucleus migration, one can envision the nucleus is either pulled by an anterior network or pushed by a posterior network. Live imaging experiments associated with laser-mediated nanosurgery ablations of MTs have shown the nucleus was mainly pushed by the MTs to reach its final destination and the reorganisation of MT network did not occur prior to nucleus migration [18,22]. Furthermore, the nucleus itself is involved in the MT network reorganisation, as is illustrated by the detection of an abnormal MT organisation in oocytes in which the nucleus fails to migrate or is mispositioned [16,42]. In fact, several MT networks have been found to be required for the nuclear migration [18]. 

## 3. Several MT Networks Participate in Nuclear Positioning

### 3.1. Centrosome Involvement

For several years, it was thought the *Drosophila* oocyte lacked active centrosomes at stages where nuclear migration occurred, as suggested by the failure to detect them during early oocyte development [49] and by the fact that the meiotic spindle formed in the absence of centrosomes at the end of oogenesis [50]. More recently, several studies have documented the existence of active centrosomes during oocyte development [16,18,24,51]. As mentioned above, the oocyte contains at least 16 centrosomes that come together to form a cluster. During mid-oogenesis, this cluster co-migrates with the nucleus and remains asymmetrically localized in close vicinity to the nucleus [16,18,22]. Importantly, centrosomes were progressively inactivated by the gradual removal of pericentriolar material during oocyte development, especially once the nucleus has migrated [24]. Several studies have shown centrosomes were responsible for the formation of MTs, which contributed to the forces that ensured nuclear movement [18,22]. The absence of centrosomes in the oocyte does not preclude the migration and antero-dorsal positioning of the nucleus as shown by 3D live imaging [18,22,52]. However, the speed of the migration is reduced and the nature of the trajectories is altered [18]. This result further illustrates migration involves redundant and/or complementary mechanisms connected to the MTs. 

The way centrosomes act as force generators for the nucleus displacement requires a specific organisation and positioning. Before stage 6, the centrosomes are relatively dispersed nearby the nucleus. A recent study has shown their clustering was a prerequisite for the centration of the nucleus (Figure 3A,B) and its subsequent migration [23]. This clustering reflects, at least in part, a reduction in centrosome activity, as it is associated with the decrease of SPD2 protein level at centrosomes. In the same line, artificially maintaining high activity of centrosomes prevents centrosome clustering and consequently nucleus centering. Finally, it has been demonstrated the microtubule-associated motor, Kinesin I, had a key role in this process.

This process involves the microtubule-associated motor Kinesin I, together with a reduction in microtubule nucleation activity, as is indicated by a decrease in the amount of SPD2 protein at centrosomes, which is essential for centrosome activity. In the absence of Kinesin I heavy or light chains, centrosomes remain dispersed and the amount of SPD2 in centrosomes is increased. Interestingly, joint inactivation of centrosomes and Kinesin I restores the ability of the nucleus to migrate [23]. In addition, another concomitant study has shown in *Drosophila* neuroblasts, the Kinesin I motor directly interacted with the DPLP protein [53] which had the ability to interact with SPD2 at the centrosomes [54]. Taken together, these results suggest a modulation of centrosome activity is required to allow their clustering and positioning between the nucleus and the posterior pole of the oocyte. 

### 3.2. A Second MT Network Operates at the Level of the Nuclear Envelope

The nuclear envelope played a central role in all the mechanisms involved in nucleus migration, often as a target for the forces exerted by the cytoskeleton [55]. In the *Drosophila* oocyte, the nucleus is a source of MTs, placing the nuclear envelope at the core of the molecular mechanism of nuclear migration. This situation is not unique as it occurs in *Drosophila* adipocytes [56,57] and in myotubes during muscle development in several animal species [58]. In the oocyte, MTs are asymmetrically organised at the nuclear envelope, with the posterior hemisphere being more enriched in MT nucleation sites. Importantly, this correlates with a similar asymmetric localisation of several proteins such as Mushroom body defect (Mud) [18,22,59], Abnormal spindle (Asp) [18], Calmodulin (Cam) [22] and the minus end MT-associated dynein motor [22,60]. Interestingly, Mud, Asp, and Cam are also involved in MT minus end focusing in meiosis II spindles [4] and their orthologs interact in the *C.elegans* oocyte [61]. In the *Drosophila* oocyte, Asp was required for the asymmetric distribution of Mud, and live 3D imaging has revealed in the absence of Mud or Asp the nucleus still migrated but at a reduced speed and with an alteration of its trajectories as it was the case in the absence of centrosomes [18]. Importantly, in the absence of both centrosomes and Mud, the nucleus fails to migrate in at least 50% of cases [18]. This indicates the centrosomes and Mud in an Asp-dependent mode are distinct molecular cues that ensure a robust MT-dependent nuclear positioning. Because nuclear migration is only partially abolished in this situation, whereas it is completely abolished upon colchicine-induced MT depolymerisation, it appears likely other MT networks are also involved in this process. Interestingly, preliminary observations suggest an additional MT network operates at the level of the posterior cortex of the oocyte (Roland-Gosselin, F., et al. [62]).

## 4. The Relationship between the Mechanical Forces Exerted on the Nucleus and Its Intranuclear Organisation and Positioning

The LInker of the Nucleoskeleton and Cytoskeleton (LINC) complex is a critical element in the transmission of forces at the level of the nuclear envelope [63]. This complex is composed of a transmembrane protein with a SUN domain (Sad1/UNC-84) that spans the inner nuclear membrane and binds, within the perinuclear space, to transmembrane proteins with a KASH domain (Klarsicht/ANC-1/SYNE Homology) that cross the outer nuclear membrane. On the nucleoplasmic side, the SUN domain protein interacts with the nuclear lamina, whereas on the cytoplasmic side the KASH domain proteins interact with cytoskeletal components including actin microfilaments and microtubules. The LINC complex could thus transmit forces from the cytoskeleton to the nuclear lamina and has been unsurprisingly involved in many model systems of nucleus migration. The *Drosophila* genome encodes two SUN proteins, only one, Klaroid (Koi), being expressed in the ovarian follicle [64,65], and two KASH proteins, Klarsicht (Klar) and Muscle-specific protein 300 kDa (Msp300). Koi, Klar, and Msp300 are present at the nuclear envelope in the oocyte [65]. 

In the germanium, during the first step of meiosis I, chromosome pairing occurs through homologous centromere pairing and centromere clustering. This process requires a rotation of the nucleus, which is mediated by the LINC Complex, together with Mud, the dynein motor, and the MTs [66]. Surprisingly, although not using live imaging techniques, a careful analysis has revealed the LINC complex as such was not essential for nuclear positioning during mid oogenesis when the nucleus is adoping ed its antero-dorsal position in the oocyte [65]. However, a double genetic inactivation of the LINC complex and Mud, or the LINC complex and Asp, impairs the migration and the antero-dorsal positioning of the nucleus (Lepesant and Guichet [67]. These results indicate the LINC complex and Mud cooperate in the positioning of the nucleus, although their precise requirement in this process remains to be established. 

Moreover at later stages, the cytoplasmic flow in the oocyte exerts forces that affect the internal organisation of the nucleus at the level of nuclear condensates such as nuclear speckles in particular [68]. Interestingly, it has been shown the chromatin landscape evolved during the development period encompassing the antero-dorsal positioning of the oocyte nucleus [69]. It would, therefore, be interesting to investigate whether the nuclear migration and the subsequent asymmetric positioning are associated with such intranuclear modifications. 

## 5. What Mechanism Triggers Nucleus Migration?

More than twenty-five years ago, two studies showed a two-step signalling event originating from the oocyte and its adjacent follicle cells triggered the migration of the nucleus [33,34]. As seen above, a Grk-mediated signal from the oocyte activates the EGF pathway in the posterior follicle cells that express the JAK/STAT pathway and triggers their differentiation into posterior follicle cells. These cells, in turn, send a signal back to the oocyte, which sets off nuclear migration concomitantly with an MT reorganisation leading to the assembly of a new MT network with the minus end at the anterior side of the oocyte [37,38,39]. The timing of these two signaling events remains to be fully elucidated, but it can be assumed they occur between stage 4 and stage 6, as the nucleus has already completed its migration by stage 7. The nature of the back signal emitted by the follicle cells remained unknown and the molecular nature of the receptor(s) in the oocyte or of the signaling cascade that caused the reorganisation of the MT network at the posterior, was still missing. This back-signal could be physical rather than molecular in nature. Although it has been observed at later stages (stages 9/10), a recent study has shown a tight contact between the apical membrane of the posterior polar follicle cells and the posterior plasma membrane of the oocyte was required to maintain MT organisation and define an anchoring zone for the *osk* mRNA at the posterior of the oocyte [70]. An attractive hypothesis that could also explain why this signal remained unknown, despite several genetic screens, would be it corresponded to changes in adhesive properties between the posterior follicular cells and the oocyte. In addition, a recent study suggests the Unc-45 myosin chaperone is required in the oocyte, as its inactivation affects the positioning of the nucleus. Of note, Myosin II, one of the targets of Unc-45 chaperone, is required in the oocyte for the posterior localisation of the polarity protein Par-1, but is not required for nuclear positioning [71]. In any case, the fact that the nucleus, once it is centered in the oocyte, exhibits an oscillatory behavior prior to its migration as revealed by live 3D imaging, suggests the nucleus can be under the influence of MT-mediated opposing forces maintaining an unstable equilibrium, that is broken by the reception of the signal emanating from the posterior follicle cells. 

## 6. Oocyte Nucleus and Meiotic Divisions

From stages 1 to 12, the oocyte remains in meiotic prophase I. During this period, the oocyte nucleus is largely transcriptionally silent, and the chromatin is compacted into a structure named karyosome [4]. Meiosis is reactivated at stage 13, with the disassembly of the nuclear envelope. At that stage, the centrosomes have disappeared and MTs have come together around the already congressed meiotic chromosomes to form the meiosis I spindle [72,73]. The stage 14 oocyte is arrested in metaphase I and remains so until ovulation triggers anaphase I. It is the passage through the oviduct, rather than fertilisation, that leads to oocyte activation and the resumption of meiosis [4]. Anaphase I is followed immediately by an entry into meiosis II. As the chromosomes move toward the spindle poles in anaphase I, the center of the spindle pinches in between the chromosomes, and an aster of microtubules forms between the separating chromosomes [73]. It is worth noting the oocyte does not extrude polar bodies; instead, all four meiotic products align perpendicularly to the dorsal anterior cortex of the oocyte [4]. The innermost meiotic product fuses with the male-pronucleus, while the remaining three female meiosis products fuse and form a single polar body. It is interesting to note, the asymmetric positioning of the nucleus results in the anterior-dorsal positioning of the four meiotic products in the future embryo, close to the micropyle, the channel through which the sperm enters.

## 7. Oocyte Nucleus Anchoring

As discussed above, the precise antero-dorsal positioning of the nucleus in the oocyte is essential for the development of the future embryo, as it controls its correct dorsal-ventral axis formation, which is critical for the establishment of the different germ layers [8]. The *grk* mRNA is tightly associated with the oocyte nucleus, and the Grk protein is translated and secreted locally at the nuclear periphery. Grk is required to specify the dorsal follicle cells by activating the EGFr, which ultimately leads to the specification of the dorsal-ventral axis of the embryo. Importantly, Grk signaling to these specific follicle cells subsets must be maintained in time, as its interruption compromises their differentiation as it happens when the nucleus is not kept anchored [39]. Another important role of the accurate anchoring of the nucleus to the antero-dorsal position in the oocyte may be to ensure after the completion of meiosis II, the four meiotic products are positioned close to the micropyle. Consistent with this, a specific morphogenetic process ensures the micropyle and the nucleus are positioned in a close vicinity during egg chamber development. One of the primordia of the micropyle channel consists of a group of somatic cells, called border cells [43] that migrate from the anterior of the egg chamber to assume a final position (see Figure 2A) controlled by Grk, whose diffusion in the oocyte is linked to the position of the nucleus [9]. In the complete absence of Grk signaling, the nucleus remains close to the posterior of the oocyte and very few embryos support embryonic development, possibly indicating they are not fertilized [37]. One might ask why the nucleus needs to be anchored at the antero-dorsal position reached when it has migrated through the dense cytoplasm of the oocyte. The oocyte, as it is often the case for large cells, is subject to an extremely high cytoplasmic streaming, which is both necessary for the development of the oocyte by ensuring cytoplasmic mixing, and important for the specification of the antero-posterior axis of the future embryo [74]. To maintain its final position, the nucleus must be anchored to resist the cytoplasmic flow. The forces exerted by this flux on the nuclear envelope are able to modulate the internal organisation of the nucleus [68]. The molecular process by which the nucleus is anchored remains elusive. Whether this anchoring is a molecular hook between the nuclear envelope and the plasma membrane or a continuous active process is currently unknown. We know, albeit fragmentarily, some of the factors required for this process to occur. The MTs are important because their depolymerisation at any time during oocyte development affects the positioning of the nucleus [16,75]. Consistent with this, it has been shown the nucleus was wrapped in a cage of MTs [16,75]. Furthermore, disruption of the minus-end directed motor Dynein or its associated cofactors Dynamitin, Lis1, and BicD, affects the positioning of the nucleus, which floats freely within the oocyte without any contact with the cell cortex when MTs are depolymerised, [60,76,77,78]. Interestingly, in some cases the nucleus is mislocalised but is still in contact with the lateral plasma membrane, such as in mutant conditions for the Kinesin Heavy Chain 1, the Phosphatidylinositol phosphate 4–5 kinase Skittles (SKTL), or the transcription factor Cap and Collar [42,60,79]. This led to the hypothesis the nucleus is differentially anchored along the anterior and lateral plasma membranes [60]. 

The Kinesin Heavy Chain 1 and dynein motors both exhibit a nuclear envelope distribution [22,23,60], consistent with an involvement in nuclear anchoring. SKTL produces phosphatidylinositol 4,5-biphosphate PIP4,5P2 present at the plasma membrane and it is implicated in MT organisation within the oocyte [79]. In mammalian cells, PIP4,5P2 is required to bind the MT-associated protein NUMA (ortholog of Mud) to the plasma membrane [80], and in *sklt* mutant oocytes, the distribution of Mud at the nuclear envelope is affected once the nucleus is migrated (Claret and Guichet [81]). This could, therefore, represent another role for Mud distribution at the nuclear envelope. Importantly, however, the exact molecular nature of this anchoring process remains to be determined. 

## 8. Conclusions

The correct positioning of the oocyte nucleus after migration at stage 7 of oogenesis is a crucial step in the life cycle of *D. melanogaster*. It is a prerequisite for the tightly localised Grk signalling required for the establishment of the dorso-ventral axis of the follicle and the subsequent determination of the three primordial germ layers in the embryo. The whole process appears to be very robust. Migration is largely, if not exclusively, microtubule-dependent and relies on redundant and separate cues based on the centrosomes and the nucleus itself. Remarkably, once nuclear positioning is achieved, it is maintained for an extended period of time throughout the completion of oogenesis until the resumption of meiosis and fertilisation. This is achieved by maintaining the nucleus in a fixed antero-dorsal position in the oocyte, despite the strong cytoplasmic flux that occurs after stage 10 and the further dumping of the contents of the nurse cells after stage 11. Beyond this stage, the nucleus is maintained in the same anterior dorsal-most position in close proximity to, if not attached to, the cytoplasmic membrane, allowing the position of the meiosis I and II spindles to be restricted close to the point of entry of the spermatozoid through the micropyle. Although it has been shown separate mechanisms mediated the attachment of the nucleus to the anterior and lateral plasma membranes of the oocyte [60], the processes underlying this uninterrupted positioning of the nucleus from stage 7 until germinal vesicle breakdown (GVBD) remained largely unexplored. Similarly, an investigation of a possible direct influence of the position of the nucleus on the dynamic changes in its internal organisation taking place during this period deserves consideration.

## Figures and Tables

**Figure 1 cells-13-00201-f001:**
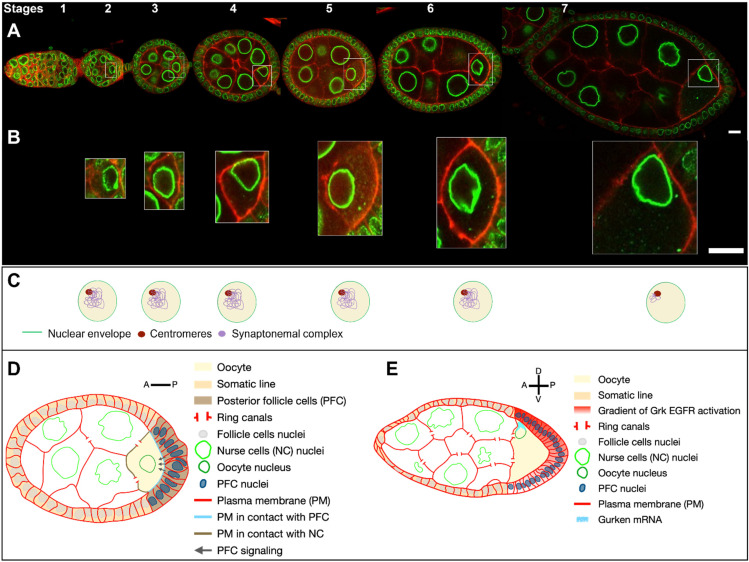
*Drosophila* egg chamber organisation and oocyte nucleus positioning from stage 1 to 7. (**A**) Reconstitution of an ovariole from different ovarian follicles from stage 1 to stage 7. The nuclei are visualized by the expression of the *Drosophila* importin-β i. e Female sterile (2) Ketel protein coupled to GFP (*Fs(2)Ket-GFP*) (green) and cell membranes by the ubiquitous expression of the pleckstrin homology (PH) domain of the Phospholipase C protein coupled to RFP (*ubi-PH-RFP*) (red) (scale bar 10 microns). (**B**) Close-up of the nucleus in the oocyte, illustrating the evolution of nuclear positioning in the oocyte from stage 2 to stage 7 (scale bar 10 microns). (**C**) Schematic representation of the internal nuclear organisation adapted from [4]. At stages 2 to 4, the chromosomes reorganise to form a compact structure of condensed inactive chromatin called the karyosome. [5,6]. From stage 5 to 6, the euchromatic synaptonemal complex is disassembled. At stage 7, the synaptonemal complex persists at the centromeres. (**D**) Schematic representation of a stage 6 ovarian follicle. The posterior follicles cells (dark brown) in contact with the plasma membrane at the posterior of the oocyte (blue dotted line) send a signal (black arrows) to the oocyte, which eventually leads to the migration of the nucleus (**B**) and the reorganisation of the microtubule networks. (**E**) Schematic representation of a stage 7 ovarian follicle. The nucleus has migrated to the intersection of the anterior and posterior plasma membranes of the oocyte. This position will define the dorsal pole of the oocyte. The nucleus is associated with *grk* mRNA, which is translated and secreted locally. This creates an activation gradient for the EGF signalling pathway, which is responsible for establishing the dorsoventral polarity of the ovarian follicle [7] and the future embryo [8].

**Figure 2 cells-13-00201-f002:**
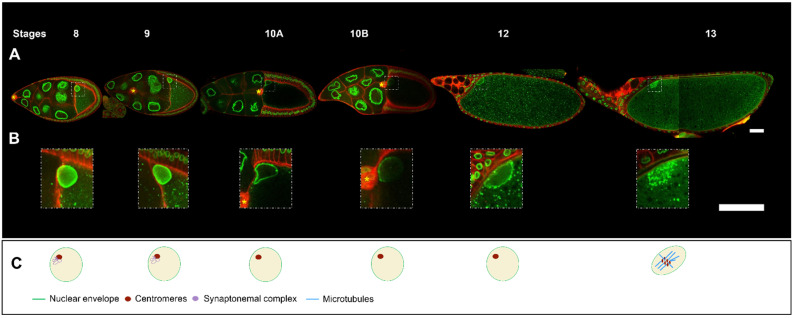
*Drosophila* egg chamber organisation and oocyte nucleus positioning from stage 8 to 13. (**A**) Reconstitution of an ovariole from different ovarian follicles from stage 8 to stage 13. The nuclei are visualized by the expression of *Fs(2)Ket-GFP* (green) and cell membranes by *ubi-PH-RFP* expression (red) (scale bar 50 microns). (**B**) Close-up of the oocyte nucleus, illustrating the evolution of nuclear positioning from stage 8 to stage 13. The border cells (yellow asterisk) which have delaminated from the anterior follicle cells at the end of stage 8, reached at stage 10 the anterior border of the oocyte and then migrated near the nucleus in the oocyte [9] (scale bar 50 microns). (**C**) Schematic representation of the intra-nuclear organisation adapted from [4]. At stages 8 and 9, as at stage 7, the centromeric synaptonemal complex persists. At stage 10, chromosomes briefly decondense and transcription is upregulated [5]. From stage 11 to 12, the chromosomes recondense. At stage 13, the germinal vesicle breaks down. Tubulin is recruited to the chromosomes and microtubules (MTs) begin to organise into a bipolar spindle. The oocyte maintains the metaphase 1 arrest configuration until it passes through the oviduct, triggering the end of meiosis I and the start of meiosis II.

**Figure 3 cells-13-00201-f003:**
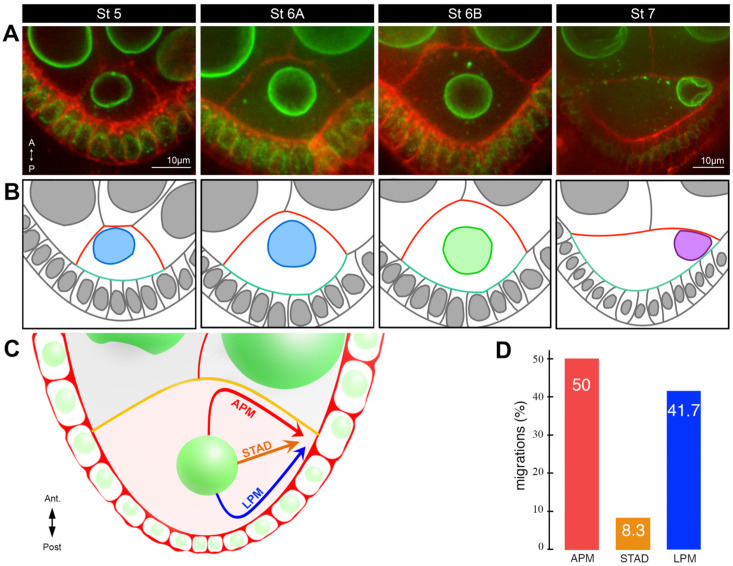
Characterisation of nucleus positioning before and during migration. (**A**) Stage 5–7 egg chambers, expressing *Fs(2)Ket-GFP* to label nuclei (green) and stained with Cellmask, a lipophilic dye to reveal plasma membranes (red). Representative examples of the different nuclear positions at stages 5, 6A, 6B, and 7 adapted from [23]. The oocytes are oriented with anterior (**A**) at the top and posterior (P) at the bottom. (**B**) Schematic representations of the image above, with a color code illustrating the evolution of the oocyte shape (anterior plasma membrane in red, posterior plasma membrane in turquoise) and the position of the nucleus: anterior, pale blue; center, pale green; migrated to the intersection between anterior plasma membrane and the lateral plasma membrane, purple adapted from [23]. (**C**) Schematic illustration of the three alternative routes by which the nucleus migrates to the antero-dorsal cortex in the oocyte: along the anterior plasma membrane (APM), through the cytoplasm (STAD), or along the lateral plasma membrane (LPM) [18]. (**D**) Bar plots of the distribution of the three different migration paths taken by the nuclei [18].

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
