# Peer review of "The Importance of the Position of the Nucleus in Drosophila Oocyte Development"

_cells, 2024, doi:10.3390/cells13020201_

Round 1

Reviewer 1 Report

Comments and Suggestions for Authors

This manuscript represents a beautiful review article summarising the current knowledge of the cell biology of oocyte nucleus positioning during oogenesis in Drosophila melanogaster. This review is written and presented by leading experts in this field and features a compact and comprehensive text with outstanding figures containing beautiful imaging data. What I liked most about the manuscript was that it was easy to read, while still referring to latest developments in the field and unpublished data. The article will be an excellent resource for experts and the interested broader readership. I have only a few suggestions that the authors may wish to address.

minor notes:

- It may be worth to explain the term 'karyosome'. From Fig. 2, the figure legend and the text referring to the figure, it was not clear what this term describes exactly.

- Typo page 3 line 84 cytosqueleton – cytoskeleton

- Page 3 line 75: nurse cells do not only synthesize mRNA for the oocyte but presumably all other RNA species, including tRNAs and rRNAs in large amounts.

- Page 6 line 165: please define what staining 'CellMask' is a trade name for a fluorescent dye. It might be worth to add a reference or other appropriate explanation. 

- Page 6 line 173: figure 3 legend should explain the abbreviations for APM STAD and LPM.

- Page 6 line 214: one 'full stop' sign too much!

- Page 7 line 247: replace ‘has showed’ with ‘has shown’

- Page 7 line 259f: replace ‘has showed’ with ‘has shown’

Comments on the Quality of English Language

The presentation in English language is excellent.

Reviewer 2 Report

Comments and Suggestions for Authors

The manuscript by Lepesant and colleagues nicely reviews a key step in the establishment of the anterior-posterior and dorso-ventral axis of Drosophila. Gurken mRNA signal is used twice to set up the posterior side and then the dorsal side of the Drosophila egg chamber.  The manuscript extensively focuses on the second signal establishing the dorsal side. The review is accurate and mentioned all of the adequate literature fairly. The manuscript is also nicely illustrated with previously published but helpful material.

I have only minor comments:

1)    Line 87, when explaining the current model for oocyte selection, the authors mention an amplification loop between microtubules and microtubule-associated motors and nucleators. To explain the model fully, the authors could mention the initial asymmetry of the fusome between the two pro-oocytes, which is proposed to bias this amplification loop toward one of the pro-oocytes. Nashchekin, D., et al #14.

2)    The authors focus on the second step of axis polarization by the migration of the oocyte nucleus: is there anything known about the localization, polarization or orientation of the nucleus during the first step of gurken signaling toward the posterior cells? At this stage, the oocyte nucleus fills in a large part of the oocyte, but there may still some mechanisms to position it.

3)    There are several unpublished data, which are mentioned in the text. I am wondering whether it is appropriate to insert unpublished data in a review manuscript?

L286; L312; L416

Comments on the Quality of English Language

the review is well written and illustrated
